# Listening to Transgender Patients and Their Providers in Non-Metropolitan Spaces: Needs, Gaps, and Patient-Provider Discrepancies

**DOI:** 10.3390/ijerph182010843

**Published:** 2021-10-15

**Authors:** Michelle Teti, L. A. Bauerband, Tyler W. Myroniuk, Erica Koegler

**Affiliations:** 1Department of Public Health, University of Missouri, Columbia, MO 65211, USA; tyler.myroniuk@health.missouri.edu; 2Department of Health Sciences, University of Missouri, Columbia, MO 65211, USA; bauerbandl@health.missouri.edu; 3School of Social Work, University of Missouri-St. Louis, St. Louis, MO 63121, USA; koeglere@umsl.edu

**Keywords:** transgender, health care providers, cultural competence, rural, non-metropolitan

## Abstract

Trans and gender non-conforming (TGNC) patients need better care; providers need TGNC focused medical trainings. TGNC health conferences can help, yet these events occur mostly in urban centers. Meanwhile, patients in non-metropolitan areas often face significant discrimination and notably poor access to TGNC care. This study explores the ongoing needs of TGNC patients and their providers following a one-day TGNC health conference in a small town in the American Midwest. Exploratory semi-structured interviews were used to gather in-depth information from TGNC conference attendees (N = 25). Theme analysis methods were used to identify areas of need for future trainings. Providers reported that they needed more exposure to TGNC patients, judgement-free opportunities to learn the basics about TGNC care, and ongoing trainings integrated into their medical school and ongoing education credits. Patients needed better access to care, more informed providers, and safer clinics. They cited lack of specialty care (e.g., mental health, surgery) as particularly problematic in a non-metropolitan setting. TGNC patients, and their providers in non-metropolitan areas, urgently need support. Patients lack specialized care and often possess greater knowledge than their health care teams; providers, in these areas, lack opportunities to work with patients and stay up to date on treatments.

## 1. Introduction


*“It’s hard to even trust that your provider needs to know what they need to know so you can just trust and be a patient, for once”*
Provider and transgender patient

Transgender and gender non-conforming (TGNC) individuals experience discrimination and poor mental health as a direct result of holding a socially stigmatized identity [1]. According to Minority Stress Theory (MST) [2,3], as a result of the discrimination and stigma, TGNC individuals experience added internal and external stressors that contribute to increased health risks. Obtaining and receiving gender affirmative healthcare services from TGNC identity-affirming health providers is related to better health among TGNC patients [4], and can buffer minority stress.

Unfortunately, TGNC individuals often lack access to affirmative healthcare, and experience added stress simply by finding and visiting a doctor [5,6]. This is especially true in non-metropolitan areas of the United States [7]. For example, a systematic review examining healthcare for rural sexual and gender (LGBT) minorities highlighted that rural providers have limited knowledge of LGBT care, negative attitudes, and that patients anticipate stigma in rural healthcare systems [8]. Another systematic review focused solely on TGNC populations similarly found that poor provider knowledge and sensitivity negatively impacted TGNC individuals’ health and recommended additional provider training to bridge this gap [9].

TGNC patients often need to educate themselves on their health in order to advocate for their healthcare, through Internet research and online communities [10]. There is a critical need for more support for providers and patients in these areas. In fact, the systematic review on rural LGBT health concluded that interventions targeting awareness, behavior, and ultimate health outcomes for rural LGBT people are critically needed [8]. TGNC healthcare conferences are a growing way to empower community members and simultaneously educate health care providers [11]. They also serve as a guide to what patients and providers need for improved care. Our research explores TGNC patients’ and providers’ ongoing training and informational needs after they attended a local TGNC healthcare conference in a small town in the American Midwest.

### 1.1. Provider Knowledge and Training

The primary challenge to affirmative TGNC healthcare access is related to provider bias and lack of knowledge [12]. Most U.S. medical school curricula do not include training healthcare providers in TGNC-specific care; only providers who seek specialized training ever receive education on how to work with TGNC patients [13]. Thus, TGNC care is viewed as a healthcare specialization; providers assume that all the needs of TGNC patients can and should be met by specialists. The perception of TGNC health as a specialization perpetuates the knowledge gap and bias among providers who choose to obtain continuing education in areas of assumed relevance to their core practice, omitting TGNC care [14].

TGNC health conferences are becoming one of the primary ways that health providers can access training in the affirmative and competent care of TGNC patients. These conferences are often held in large cities and urban centers, where TGNC community and health clinics are more accessible (e.g., Mazzoni Center, 2021; National LGBTQIA+ Health Education Center, 2021), and TGNC individuals tend to have more social protections. Although some of these conferences are free for community members to attend, there are still significant costs to both providers and community members to travel and seek lodging for such events. For most TGNC community members who live outside these more affirming metropolitan areas, TGNC health conferences are less attainable, and they primarily seek information through online resources [15]. Additionally, the requirements of travel and taking time off work make these health conferences less desirable for continuing education among healthcare providers—especially those who must finance their own training.

Moreover, the needs of providers and TGNC patients in urban and rural or small-town settings differ from those in urban settings [7], and arguably, those in smaller towns need more support. Smaller communities are less diverse, overall, than urban communities, and stigma, confidentiality challenges, and limited services are more significant factors in non-metropolitan areas than larger cities [16]. The greater homophobia and discrimination experienced by LGBT people living in non-metropolitan areas has long been known [17], and such experiences vary by geographic region for LGBT [18] and TGNC individuals [19]. For example, discrimination is higher in Southern and Midwestern states versus the Northeast or Western United States [20]. Similarly, Sinnard et al. (2016) found correlations between anxiety and depression and geographic location for TGNC individuals in Midwest regions. In addition to these challenges, fewer healthcare providers serve TGNC people, and those who do may not have the same expertise as in metropolitan areas.

### 1.2. Diversity of TGNC Individuals

TGNC individuals are diverse in terms of their gender and TGNC-specific healthcare needs. Research on TGNC healthcare has historically focused on transition-related needs, specifically around accessing hormone therapy and gender-affirming surgeries, although not all TGNC individuals seek the same medical services, even if they share similar gender identities [21]. Even when providers in rural areas are willing to work with TGNC patients, their awareness and knowledge of diversity within the population may be limited. A qualitative study of TGNC healthcare experiences in rural areas described the specific issues with having limited providers who work with TGNC people, including increased costs to visit those providers and facing difficulty receiving non-transition specific healthcare [22]. Given the diversity of this population, and the required skills needed to be TGNC-affirming as a provider, there is a need to explore how TGNC healthcare conferences are received in a non-metropolitan area.

### 1.3. Local TGNC Health Conference

The purpose of the current research was to explore the unmet medical training needs among providers and their TGNC patients after they attended a TGNC health conference in a small town in the middle of the United States, to fill gaps in research about the needs of TGNC patients and their providers in non-metro settings and inform future training.

## 2. Materials and Methods

### 2.1. Setting

Study participants included attendees at the Summit to Improve Transgender Collaborative Healthcare (STITCH). STITCH was an all-day training event targeting both patients who identified as TGNC and providers who served or wanted to learn more about serving TGNC patients. The intended provider audience included practicing clinicians, health educators, and health care students. The event sought to educate the attendees about standards of transgender care; how to manage medical, surgical, and mental health care for transgender patients; foster collaboration across healthcare disciplines on patient-centered treatment plans; create inclusive spaces and improve clinic policies to help meet trans patient’s health needs; and describe the rights of transgender patients. STITCH was widely publicized through multiple list-servs and medical and community spaces. An estimated 176 people attended the event. Participants could choose to attend patient- or provider-centered discussions from various tracks (e.g., community, mental health, and medical) offered throughout the day.

### 2.2. Study Participants

After the event, participants (e.g., providers and patients) completed an evaluation of the sessions attended and overall experience of the event. Event participants were asked if they would be willing to participate in a follow-up interview to help evaluate the effectiveness and overall experience of the event. Using this list, we emailed all providers who said they were providing some form of physical patient care (i.e., not just studying care and/or a medical student) and all patients who had received physical care during the last year.

We omitted practitioners who reported providing only mental health care (e.g., counselors, psychologists) and patients who had not sought care for physical health issues, because, given the small sample appropriate for qualitative work, we wanted to focus the discussions on care as much as possible. We recognize that physical and mental health concerns and care are associated, but by omitting providers and patients who only provided or sought mental health services, we narrowed the scope of interview issues to sufficiently gather more in-depth data, versus more general information. We aimed to interview around 10 to 15 individuals from each group and revisit sampling if saturation or repeated themes were not achieved. We interviewed 25 participants—13 providers and 12 patients—which proved to be a sufficient sample.

The group planning the conference agreed to having us conduct post-conference interviews if the research team did not collect demographic information about individual participants. The conference took place in a very small town; therefore, the planning group was concerned that even minor demographic descriptors could identify participants. Thus, we only have general information about the samples. Providers identified as men and women in equal amounts and practiced in a state in the middle of the United States serving both rural and urban populations. They were medical residents, primary care physicians, midwives or obstetrics providers, or nurses who provided primarily physical health care services. Providers’ ages varied between 20 and 60 years and the majority of the providers were White. Patients who identified as TGNC included an equal amount of male, female, and non-binary gender identities. Patient ages ranged from 20 to 40 years old, and the majority of the sample identified as White. Patients resided in the same state as practitioners. All patients had sought some form of physical health care in the last year.

### 2.3. Procedures

All research activities were approved by the first author’s IRB. Semi-structured interviews took place by phone, were recorded, and lasted around 30 to 45 min. Interview questions for both groups inquired about conference experiences, lessons learned, the most important content, and training needs. The interview guide is in Appendix A.

### 2.4. Analysis

We used a theme analysis approach to identify key patterns in the data [23]. Guest outlines a protocol by which researchers review, discuss, code, and interpret the data in an iterative process. A team of three researchers—the first and second authors and a research assistant—conducted all interviews and had basic familiarity with the data. As per Guest, once the interviews were transcribed, these three researchers reread all the data and independently generated a list of key themes that came up in the data around the conference. Next, the group met to compare their lists and create a master list of themes for further exploration in the data. Interview themes broadly included the most important things learned, remaining gaps in learning, and the conference experience. Next, we decided to focus this analysis on remaining gaps in learning, to shed light on the future direction of training for providers and patients, especially in rural or non-metropolitan areas. We created a codebook out of key themes related to gaps in learning for each group (providers, patients) that defined each theme. Then, together, we analyzed a set of three transcripts using the codebook as a guide to match the text to corresponding themes. Subsequently, we met and discussed coding, clarifying discrepancies and the codebook, until a sufficient level of agreement was reached on the coding scheme. Next, the research assistant coded the remainder of the data. We met as a team throughout the analysis to discuss themes, connections between themes, and notable findings. Lastly, we created a code report that listed all data by theme and used this to write the study’s results.

## 3. Results

### 3.1. Providers

Three themes were prominent among providers. When discussing their experiences at the conference, they identified ongoing unmet needs around: (1) seeing and being seen; (2) understanding the basics without judgement; and (3) integrated, in-depth, longer-term training.

***See and Be Seen.*** Providers described little access TGNC patients, and thus, few opportunities to know and understand this group. When describing their needs, providers often lamented things such as, “[I don’t see] any level of transgender patients where I work,” or “I want to be more involved [with this population].” They believed that this lack of access hindered their learning or ability to put what they learned to use. Thus, they expressed a need for patient perspectives, stories, and examples. As one provider explained:


*“I really liked the ‘true to life ‘experiences that a lot of the trans patients talked about. I thought that was very powerful and for someone who doesn’t work in the trans community every day. I think that is an excellent learning tool, to get that feedback.”*


Similarly, other providers said that these stories were critical: “I don’t think [trans health] is something we talk about normally in our everyday lives unless we know someone” and “I appreciated the personal experiences and [hearing] about things that I never considered as something to be concerned with as a medical professional but very much are.” Summing up the thoughts of many, a provider remarked that unless she was able to “be with” TGNC patients, she could not put the information into practice.

Providers also assumed that they needed to be seen by potential trans patients to be trustworthy, a belief that was not supported by the patient-expressed needs (see the next section). Nonetheless, providers said it was important to “rub elbows with all the people in the audience that are trans” and put their name and face out to the community as a “safe way to enter the healthcare system.” One said it was important for TGNC patients to “see that they have support.” She said she tried to talk to as many people as she could to build trust. Another explained, “I think they feel comfortable to know that there are people who want to offer care to them.” Yet another provider elaborated on this point, remarking:


*“I think trans people coming to the conferences and seeing how many health professionals really care about this and really want to learn more about this I think might kind of help them see that, you know, like, it’s not always going to be a negative experience, and that it’s something that we health professionals are really, like, trying to work on so that we can provide better care for them. So, I think it would be nice to have that interaction between trans patients and health professionals.”*


***Understanding the basics.*** Providers lacked a basic level of understanding of terms regarding TGNC people and their care and needed a safe space to ask questions without judgement. Pat said the conference just kept reminding her “how much I need to learn.” Regarding how to care for TGNC patients, one provider noted that she hoped, at one point, that providers would have an open mind and at least know who to turn to for the right information but “we are obviously not there yet.” Several providers had questions about pronouns and the definitions of gender terms (non-binary, etc.) and how to address these things in patient interviewing and on medical forms. One provider said she learned about pronouns and immediately put it to use in a Grand Rounds presentation. Another provider realized that their clinic was not paying attention to names when they called out to patients in the waiting room.

Providers were eager for information on how to improve in these areas. One noted, “talk to me like I’m dumb…give me a list of resources.” Along these lines, another said:


*“I would have loved to have had somebody who has already created within their clinic some real gender-neutral or gender-inclusive ‘languag-ing’. Like I would have loved if someone said, ‘So our doors say this. Our actions say this. Our intake form says this. Our people that answer the phones say this.’ Like, I would have written it down verbatim and used it…I don’t need to reinvent the wheel. I just want you to tell me what your wheel looks like, and I’ll copy it.”*


***Integrated comprehensive training.*** Given that terms were a challenge, it is not surprising that most providers did not know what they needed or wanted to know about hormones, surgery, and post-surgical care. One stated that she did not know “that type of surgery was possible.” Providers wanted to help all their patients, however, and struggled because, as Julie noted, “medical providers don’t think outside of the gender norm box that they’re trained in.” Providers said they preferred this training to be incorporated into their past and ongoing medical training. They believed that this would make training more accessible to more providers, would reach providers who may not realize that they needed it, and would allow for more complex and in-depth training.

Almost all providers noted that they lacked TGNC training in the medical curricula, or that the training was brief or inadequate. For instance, one provider noted that their training was an “all-encompassing queer LGBTQ lecture on adolescent health. And it was just about giving patient interviews, but nothing specific and definitely nothing nearly as in-depth.” Some medical students were not sure if or when trans content would be covered, and one provider noted that it was too late because she was already volunteering in a “gender-affirming clinic [but I am] kind of clueless about everything.”

One participant, who was a provider and also someone who identified as transgender, said that patients could not even trust that their doctors understood their concerns and had to educate and advocate for themselves, and they could not “just be a patient.” Confirming this, many providers said they were not sure how to handle topics such as hormones or surgery. One provider noted that prior to the conference, he was “completely clueless” about where to obtain information about these issues for patients. Despite patients wanting providers to be knowledgeable on all topics (see the next section) some providers, even after the conference, did not have the confidence to prescribe hormones. As one provider explained:


*“I am ready to see patients for other endocrine problems, but I am not doing transgender care. I don’t think I am trained enough or have learned enough to do transgender care. So, I want that to be done by people who have a special interest and are capable.”*


### 3.2. Patients

Among patients, themes centered around needing to overcome barriers related to (1) accessing care; (2) providers and clinics; and (3) accessing specialized needs—especially locally.

***Accessing trans care.*** A few participants gave examples of experiences when they could not access care. For example, one patient said they went to a clinic and was told, simply “We don’t do trans stuff here.” Another patient said they found that they could access care at a local Planned Parenthood but was challenged by the “people outside who yell at you and stuff.” A few people also mentioned basic access barriers such as transportation.

Most commonly, many patients faced financial and insurance barriers. Lacking the ability to pay for doctor visits, hormones or surgeries—as well as other transition components such as voice modification or name changes—was the primary reason people were not making these changes. A few people mentioned insurance as a top health concern. Another patient explained that an additional session on insurance would improve the conference:


*“Bring in somebody who is an expert in insurance who can talk about policies. Because that seems to be where most of the folks I know run into difficulties–getting a straight answer from their insurance provider. They tell you one thing and then you work on that for a while, and then, when the time comes [for a procedure], they’re like, ‘Oh, no; that’s not the thing you’re supposed to do.’”*


A different patient explained that several states had grants for confirmation surgeries, and that this was important and in need of expansion. Sa said that the most important thing they learned at the conference was a potential way to access the services they wanted covered:


*“Panera Bread covers facial feminization surgery. So now my dream job is to get a job at Panera Bread eventually. Because that’s the surgery I need the most, because that’s the one I think will affect my life in the most positive ways and help my mental wellbeing. That one’s not covered by insurance in 99.999 percent of cases.”*


***Challenges with Providers.*** Patients noted the need for less biased and better-educated providers. Most of the interviewees noted that they wanted to be treated “like a person, instead of a big stigma”, or “a whole person” or “a human being.” One patient described being dismissed by doctors, which worried them, particularly given “how the medical profession has objectified people with bodies like mine and identities like mine.” Thus, they called for readily available lists of providers so they could identify supportive doctors and avoid biased or uneducated providers. One person admitted to staying with their pediatrician because “I know I can be comfortable with her, and I don’t want to lose the person I made this good connection with and who will be safe.”

Although providers believed it was very important to show their face (see the above section), patients wanted more than that. One patient explained that their doctor was “nice and all” and told them, “I’ll do whatever you want.” Although they appreciated the intention, the same patient said, “I don’t know what I am doing though, can’t you help?” Providers acknowledged that many patients also felt pressured by and did not like having to “teach their provider as they build a relationship and help them understand what is acceptable and not.”

Patients noted that at the conference they were surprised that providers attended the “community track” sessions, which many trans patients assumed were for patients only. TGNC want ongoing TNGC learning spaces. One patient explained, “community sessions could be closed to the community…for the community to gain access to resources…it was disappointing that these were dominated by health care providers” who had basic questions. Someone else noted that these sessions became watered down and “lacked detail.” Thus, although providers needed and wanted this basic information, patients craved detailed and in-depth information. Patient’s goals were to “check items off on [my] transition list.”

***Specialized Needs*.** On this note, a final theme describing patients’ needs is their need for local specialized care. Some participants talked about hormones, but the more predominant unmet specific medical needs were mental health care and surgery. Most participants mentioned both the importance of mental health and the lack of local mental health care. As one person put it, “it’s very difficult to find mental health care…providers who get it…and are safe enough…I know a couple families that travel [2 h away to urban area] to find a therapist good with youth and trans stuff.” Another patient stated that finding a mental health professional that was “both affirming and willing to take insurance” was very hard. A different patient called the absence of local mental health care the biggest local barrier. EK explained that the combination of being trans and anything else—including mental health care, having a disability, having a specific physical problem—seemed to be very difficult for providers to support.

One respondent pointed out that having community and mental health support were key to preventing trans people’s anxiety, depression, and suicide—making non-metropolitan areas, such as the site of this conference, very risky environments. Sa described mental health as the most important immediate need for trans people:


*“When someone comes out as trans, and they make the decision to transition-their life is going to go through hormonal changes, social changes. They’re going to lose family and friends. They’re going to make new family and friends. People try to revolve around homeostasis… but all these changes have to occur. You wouldn’t be at that point to transition if you weren’t suffering from some sort of dysphoria in your life, whether it has to do with your gender or something.”*


***Lack of local resources in the area were harmful and disappointing for trans patients.*** The majority of patients said that the conference presentation on surgery was the most beneficial session because of the detailed surgical information, including where to receive surgery within a day’s car drive. For example, one patient participant said they took “seven or eight pages of notes” during this session. Several people mentioned receiving poor previous surgery advice from doctors who acted as gatekeepers, incorrectly telling trans people that surgeries needed to be performed in a certain order, which deterred them from seeking more information. Beyond information, finding a place to have and recover from surgery was a major unmet need. Living in a place with less access to talk to surgeons or people who had undergone surgery, one patient said, “there are so many terrible stories out there that you get nervous…But to see people that had top or bottom surgery [in the presentation] be so happy after…seeing those images of people being so happy was really incredible.”

Ant explained their concerns about accessible local surgery options:


*“Part of me wanted to see if I needed to plan to go halfway across the nation, or whether it may be a possibility to have [surgery] around here. I remember coming home and telling my mom, like, ‘Hey, there’s a place that’s really close. We could just drive and come back, and I would be able to recover where I’m comfortable.’ And then now it’s just a matter of having the money ready and making the appointment. So that’s really good to know.”*


Another patient said they wanted someone to “take their hand” and give them “assurance that [local town] is moving forward.” Although they did not hear that exactly, their needs around surgery resources were partially met.

## 4. Discussion

The evidence is clear that TGNC patients need better health care and health care experiences [24,25], and providers need more information and tools to help these patients [26,27]. As shown in our research, trans health conferences and safe environments where patients and providers can meet and learn are crucial in beginning to align providers’ practice with patient needs. Given the lack of trans health care in non-metropolitan areas and the larger obstacles TGNC face to care in such settings, conferences are particularly important for smaller towns [28].

Attendees at our conference enjoyed networking and learning from one another. However, each group expressed ongoing needs after the conference that shed light on helpful ongoing training opportunities in conferences and other settings for TGNC and providers in non-metropolitan areas. Patients had ongoing needs accessing care and knowledgeable, high-quality providers, and ensuring that more complex health needs, such as surgery, were met locally. Providers wanted more experiences with TGNC patients, opportunities to learn the basics without judgement, and the ability to integrate this learning into their ongoing training and practice.

### 4.1. Key Patient Findings

Our findings support existing research that highlights the importance of improved access to care and skilled providers to promote TGNC people’s health [29]. Our findings push this research forward by identifying the need for this care to be met locally, as well as the implications of not having it accessible across geographic jurisdictions—particularly areas outside of large metropolitan areas—leading to potentially adverse outcomes for patients. Individuals living in distant, smaller towns, seeking a facial feminization surgery, for example, are inevitably burdened by the costs of travel and lodging, on top of the already physically and mentally stressful circumstances. Patients might then be without optimal post-surgery care in their home jurisdiction. Some patients also described the inaccessibility of mental health care too. Local providers’ inability to make referrals and answer complex questions for TGNC patients is also problematic—a fundamental issue not overcome by providers simply being accepting and kind. In summary, our findings showed major areas of need for TGNC patients in a small, Midwest town.

### 4.2. Key Provider Findings

Providers in our analytic sample signed up for the trans conference and were individuals who wanted to learn about TGNC health—not a general population sample of health care providers. Not surprisingly, stigma towards TGNC people was low; however, surprisingly, knowledge about their health care needs was lower than anticipated. Providers said they had trouble gaining experience without having TGNC patients in their practices and, therefore, basic information such as terminology (e.g., transgender, gender non-binary) was confusing to providers. Providers also admitted that they were afraid to ask questions for fear of judgement, leaving fundamental questions unanswered. To address this problem, providers suggested that they needed more opportunities to hear stories and experiences from TGNC patients. They did not, however, note the potential implications of this idea for patients. For example, providers did not discuss the burden this might put on TGNC patients or the stigma or risk of them telling their stories in public forums. Similarly, providers said they wanted to be seen by TGNC patients with the assumption that they would be labeled as “safe”, although there was no evidence on the side of patients that this visibility alone was verifiably helpful. Providers also thought that ongoing education was the most helpful form of training for them, but said that they lacked these opportunities.

### 4.3. Implications for Practice

***Patient Care***. To protect themselves, our findings suggest that patients can utilize their networks to identify more affirming local providers and institutions that accommodate various insurance situations, given the reality of the availability of affordable care and expertise in some small-town locations. Researching TGNC-friendly care on existing websites and databases may also help direct care-seeking. Once in the appointment, patients can be prepared with a list of items and questions. If a doctor is not familiar with a TGNC-related question, the patient may need to ask to see another doctor or ask the doctor to conduct additional research. Our findings also suggest that building patient confidence, advocacy, and empowerment skills may assist them in their journey to find appropriate care, similar to the ways in which people living with HIV in sub-Saharan Africa have done when health care providers have lagged far behind understanding and addressing their medical needs [30]

***Provider Needs***. Our findings indicate that providers, especially in non-metropolitan areas, may need basic information on TGNC individuals and the care they need, but may be afraid to ask for it; health care provider training may need to occur on multiple levels—ranging from basic to advanced—to fill a multitude of provider needs. This is consistent with other middle-America TGNC provider research, which indicated a necessary shift from cultural to clinical competence via a tiered competency model [31]. Ongoing training and training events that provide access to the patient experience would be ideal, according to the providers in this study. This means that partnerships, perhaps between academic, clinical, and community sites, may be helpful to provide ongoing support. Patient panels can be a burden or stigmatizing; therefore, storytelling models of intervention—where the stories or experiences of TGNC people who want to share their stories can be highlighted or recorded—may be worth further study [32].

***Structural Solutions***. Beyond patient empowerment and provider information, structural changes may be necessary, especially in locations where TGNC patient needs are not being met. Ideally, training built into medical schools or continuing medical education would help providers stay abreast of these issues and may help reach providers who do not volunteer for such trainings. If providers are simply unable to address questions on-site, it is possible that telehealth models, such as those used for HIV care to transition care from a specialty to primary care [33], may also provide additional training and support. Patients need access to more complex care, information, and referrals. Insurance programs need to adapt to cover treatments that TGNC require to be healthy [24]. If these treatments are not covered, the financial gap between members of the TGNC community who can afford and cannot afford care will continue.

***Suggestions for Future Research*.** This study, in alignment with Rosenkrantz et al. [8], suggests that more intervention research is needed to enhance positive health outcomes for TGNC people, especially in small towns and rural locations. Institutions that conduct conferences and workshops, such as the one presented here, on a national scale, should include evaluation and dissemination of their findings to enable shared learning. Additional research should explore and establish the most effective curriculum to train providers to deliver affirmative TGNC in rural locations. Finally, rigorous longitudinal evaluations are needed to determine whether training translates to improved health outcomes. In addition, providers in non-urban settings may feel more isolation around working with TGNC patients. Single-event conferences provide an accessible avenue for continuing education and motivation to work with TGNC patients; however, future research that explores the efficacy of provider support networks, especially for physicians learning to care for TGNC patients, is needed.

### 4.4. Limitations

The sample only represents people who want to learn; thus, participating providers may already have reduced levels of stigma towards TGNC patients compared to providers who do not seek and attend these types of training events. Similarly, TGNC patient participants were those willing to engage in such a forum, implying a strong sense of comfort in discussing their needs and medical issues with providers with whom they were not familiar. Additionally, many of the more senior medical providers who attended opted out of follow-up interviews, and even those who did volunteer to be contacted did not always respond. This suggests that the busiest or most advanced TGNC providers did not provide feedback on the conference experience or their needs. Consequently, the sample was biased, in unknown ways, by including the most active and engaged providers and patients, although being unable to acquire the perspectives of reluctant, apathetic, and/or unaware health care providers and TGNC patients who were uncomfortable with this conference format. Lastly, the findings are likely most relevant to providers and patients in small Midwest towns, but it is unclear the extent to which these findings may be applicable to small towns outside the Midwest, or more generally, to the rest of the United States.

## 5. Conclusions

Although improving TGNC people’s care is becoming a more well-recognized goal, nationally, geographic disparities in care elucidate the work that needs to be done. Outside of large metropolitan areas, the disproportionate lack of access to health care facilities and providers relative to population size, the lack of provider training, and social stigma—often due to unfounded religious and political beliefs—are well-known hindrances to achieving this goal; the experiences and needs of health care providers and TGNC patients along the rural–urban continuum can be dramatically different. Practically, it is difficult for health care providers and patients, and academic scholarship, to convince private companies to invest in health care facilities, or to expect deeply ingrained stigma among the general public, in less densely populated areas. However, the findings from this paper indicate that continuing and expanding health care provider training, with strong input from TGNC individuals, would produce the most impact on improving the care and well-being of TGNC patients; this is within the control of practitioners and scholars.

## Data Availability

Due to the privacy of participants, these data are unavailable to external researchers.

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
