# Peer review of "Listening to Transgender Patients and Their Providers in Non-Metropolitan Spaces: Needs, Gaps, and Patient-Provider Discrepancies"

_ijerph, 2021, doi:10.3390/ijerph182010843_

Round 1
Reviewer 1 Report
The authors present a manuscript on client/healthcare providers perspectives on trans* individual healthcare in rural US American contexts.
Overall, I believe the paper is well written and enriches the picture on trans* healthcare needs, and the lack of services in non-specialized or non-metropolitan areas (of the US, and elsewhere). I do think that it would be possible to derive more specific research questions based on the literature that the authors are citing, and not be as "exploratory" as they make it sound.
Methods: The interview questions should be posted in an Appendix. Now it is difficult to assess what has been actually asked. Also the authors should describe their analysis strategy in more detail; and how the material was coded (by how many coders, etc).
Participants: More information about the participants should be provided. What were the specialisations/positions of the healthcare providers, how many years have they been working in the field, ideally also location of training etc. The same should be provided for clients, at least age, if possible some more information about their trans*care needs experiences
Results: I find it unusual that the quotes are woven into the text and presented narratively. Usually, larger sets of statements and sentences are presented to highlight an interpretation provided before or after.
Discussion/Limitations: The authors should make it more clear to which context this pertains (US mid-west only), and not make it sound that these findings are universally applicable; also the authors should talk about limitations based on the selection of questions asked (given that the interviews were relatively short). In general I would present these finding in a more "humble" way.
Author Response
See attached document, thank you!

Reviewer 2 Report
I appreciate very much the topic studied in this article. Overall, the manuscript is well written. However, the manuscript has several weaknesses that must be improved.
Introduction lacks depth, theoretical relevant terms should be included and discussed, and preferably a theoretical model that helps to explain the phenomeno.
More details regarding the procedure is valuable, including a table or scheme with the structure of interviews. Is there any qualitative procedure or protocol that the authors could describe (for example, they refer to Guest et al., 2012)?.
Regarding limitations of the present research, Discussion should provide suggestions for future research. To my view, the section Implications for practice goes beyong the study and conclusions, please be careful about it.
Author Response
Please see attached letter, thank you!

Round 2
Reviewer 2 Report
The new version of the article is acceptable for publication. Details provided have considerably improved the manuscript. However, authors should complete the following sections: Supplementary material, Author contributions, Funding, Institutional Review Board Statement, Informed consent Statement (etc.).
Author Response
Thank you for your suggestions. We added the appropriate "back matter" - IRB, conflict of interest, etc. - as suggested.
The authors